# A Conservative Combined Laser Cryoimmunotherapy Treatment vs. Surgical Excision for Basal Cell Carcinoma

**DOI:** 10.3390/jcm11123439

**Published:** 2022-06-15

**Authors:** Lucian G. Scurtu, Marian Petrica, Mariana Grigore, Alina Avram, Ionel Popescu, Olga Simionescu

**Affiliations:** 1Department of Dermatology I, Colentina Hospital, “Carol Davila” University of Medicine and Pharmacy, 19-21 Stefan Cel Mare Road, 020125 Bucharest, Romania; lucian.scurtu@drd.umfcd.ro (L.G.S.); mariana.grigore@umfcd.ro (M.G.); alina.avram@umfcd.ro (A.A.); 2Faculty of Mathematics and Computer Science, University of Bucharest, 010014 Bucharest, Romania; marianpetrica11@gmail.com (M.P.); ioionel@gmail.com (I.P.); 3Institute of Mathematical Statistics and Applied Mathematics of the Romanian Academy, 050711 Bucharest, Romania; 4Institute of Mathematics of the Romanian Academy, 010702 Bucharest, Romania

**Keywords:** basal cell carcinoma, imiquimod, CO_2_ laser, 5-fluorouracil, cryosurgery, cryoimmunotherapy

## Abstract

Surgical excision is the standard treatment for basal cell carcinoma (BCC), but it can be challenging in elderly patients and patients with comorbidities. The non-surgical guidelines procedures are usually regarded as monotherapy options. This quasi-experimental, non-randomized, comparative effectiveness study aims to evaluate the efficacy of a combined, conservative, non-surgical BCC treatment, and compare it to standard surgical excision. Patients with primary, non-ulcerated, histopathologically confirmed BCCs were divided into a conservative treatment (129 patients) and a standard surgery subgroup (50 patients). The conservative treatment consisted of ablative CO_2_ laser, cryosurgery, topical occlusive 5-fluorouracil, and imiquimod. The follow-up examinations were performed 3 months after remission, then every 3 to 6 months, and were extended with telephone follow-ups. Cosmetic-self assessment was recorded during a telephone follow-up. Subjects from the conservative subgroup presented a clearance rate of 99.11%, and a recurrence rate of 0.98%. No recurrences were recorded in the surgical group, nor during the telephone follow-up. There were no differences regarding adverse events (*p* > 0.05). A superior self-assessment cosmetic outcome was obtained using the conservative method (*p* < 0.001). This conservative treatment is suitable for elders and patients with comorbidities, is not inferior to surgery in terms of clearance, relapses, or local adverse events, and displays superior cosmetic outcomes.

## 1. Introduction

Basal cell carcinoma (BCC) is the most common skin cancer in Caucasians, mainly diagnosed in fair-skinned individuals. BCC accounts for 80% of skin cancers, and one-third of all cancers in Western countries. This monomorphic neoplastic proliferation of the basal keratinocytes has a mild biological behavior, and is a slow-growing, non-aggressive cancer. BCC does not have precursors, and its rate of metastasis varies between 0.0028% and 0.55% [1,2,3,4]. However, BCC can be locally invasive, depending on its histological subtypes: low-risk (superficial, nodular, pigmented) and high-risk (morpheiform, infiltrative, micronodular, and basosquamous) [5].

BCC treatments are standardized through international guidelines that take into account the histological subtypes: standard surgical excision, Mohs micrographic surgery, radiotherapy, photodynamic therapy, curettage, electrocautery, cryosurgery, laser ablation, topical treatment with imiquimod, 5-fluorouracil, and oral Hedgehog signaling pathway inhibitors. The choice of treatment depends on tumoral features (localization, size, histological subtypes, recurrence risk, and importance of tissue preservation), patient-related factors (age, comorbidities, preferences), doctors’ training, and surgical skills. Mohs and slow Mohs micrographic surgery are not widely available, and require intensive training.

Radical surgical excision may also be challenging in elderly patients and patients with comorbidities in terms of anesthesia safety. On the other hand, the non-surgical guidelines procedures are usually regarded as monotherapy options. They include imiquimod or 5-fluorouracil, and tissue destructive options, such as curettage, electrocautery, cryosurgery, laser ablation, and photodynamic therapy [6,7,8,9,10,11,12]. Since not all dermatology departments have operating theatres, conservative treatments may be more suitable in certain circumstances. Cosmetic concerns represent an attractive feature of conservative treatments, especially for less approachable areas (e.g., auricle). 5-fluorouracil is an antitumor compound whose local metabolites interfere with DNA synthesis and RNA functions. Furthermore, it increases p53 expression, and provides a potent antitumor effect by inducing apoptosis in cancerous and precancerous skin tumors, such as actinic keratosis, BCC, squamous cell carcinoma, and keratoacanthoma. As it penetrates only up to 1 mm into the skin, the low local absorption of 5-fluorouracil may represent an important limiting factor [13,14]. 5-fluorouracil chemowraps (under occlusive bandages) may overcome this inconvenience, as was shown in squamous cell carcinomas and actinic keratoses [15]; the combination with cryosurgery was found to be efficient in treating BCCs [16]. Imiquimod is a guanosine analog that stimulates innate immunity, and provides an antitumor effect. It is approved for the treatment of BCCs and other skin cancers, as well as cutaneous inflammatory disorders [17]. Classic immunocryosurgery comprises imiquimod applications and cryosurgery. While imiquimod stimulates local activation of potent antigen-presenting cells, the cryosurgical insult promotes necrosis of the residual tumoral cells [18]. The novel EADO classification for the BCC provides a five-group classification of difficult-to-treat BCCs based on five patterns of clinical circumstances [19]; future treatments should be guided by this current position statement.

Nevertheless, current knowledge about the outcomes of a combined CO_2_ laser, cryosurgery, imiquimod, and 5-fluorouracil is not sufficiently vast. The aims of this study are: (1) to deliver an original and manageable conservative treatment (CTr); and (2) to compare it with standard BCC surgery in terms of outcome, adverse events (AE), and cosmetic outcomes. This sequential treatment should be suitable for ambulatory care, tailored to elderly patients, patients with multiple comorbidities, or patients who refuse surgery.

## 2. Materials and Methods

### 2.1. Study Design and Patient Recruitment

A total of 179 patients with BCC from two academic centers were enrolled prospectively between 1 January 2013 and 15 December 2018 in a comparative effectiveness research study. The study includes adult patients (≥18 years) having biopsy-confirmed primary BCC. All patients provided informed consent, and the study was conducted in accordance with the ethical principles defined by the Helsinki Declaration, and was approved by the Ethics Committee of Carol Davila University of Medicine and Pharmacy (PO-35-F-03), Bucharest, Romania. Patients were divided based on their treatment choice, into 2 subgroups for evaluation: conservative treatment (*n* = 129) and standard surgical excision (*n* = 50). Patients with Gorlin syndrome, recurrent BCC, ulcerated BCC, relapsed BCC, patients with a history of severe immunosuppression (chronic uncontrolled infections, chemotherapy, radiotherapy, other immunosuppressive therapies) in the last 5 years, and pregnant patients were not included in this study.

### 2.2. Procedures

The following parameters were recorded initially in all patients: clinical examination details, digital dermoscopy, medical history, sex, age, skin Fitzpatrick’s phototype [20], localization, size, histological BCC subtype, and the reason to undergo one of the two treatment methods (CTr or radical surgery). The anatomical sites of BCC were classified as follows: nose, neck, ears, cheeks, frontotemporal, scalp, limbs, and trunk, respectively. The BCC subtypes were classified as superficial (S), nodular (*n*), and others (O) (this included patients with pigmented variants and sclerosing BCCs). Before treatment, the standardized BCC dermoscopy criteria were applied [21]. The same criteria were used for the post-procedural evaluations.

#### 2.2.1. Conservative Treatment

A punch biopsy was performed prior to the CTr for diagnosis confirmation. The CTr consisted of a quasi-experimental protocol comprising ablative CO_2_ laser, cryosurgery, and topical 5-fluorouracil and imiquimod. The novelty of this protocol consists of the combination of the otherwise monotherapeutic options into a multi-step conservative approach.

The CTr (Figure 1) started with a full ablative CO_2_ laser therapy and laser hemostasis under a topical anesthesia of lidocaine 2.5% and prilocaine 2.5%. For this, 1–3 laser passes with the power adjusted to 5 to 7 W/cm^2^ (beam diameter 1mm) were used, with a security clinical margin of 5 mm around the suspected area. The angiogenic component guided the laser depth, and indicated the elimination of the tumor. Laser ablation was followed by a 5-fluorouracil (5FU) chemowrap for 24 h. Chemowraps consisted of 5% 5-FU cream to the skin under a compressive self-adhering bandage. Following 5FU chemowrap removal, an antibiotic powder containing zinc bacitracin 250 UI and neomycin sulfate 5000UI was applied for 5 days by the patients, twice a day. After a 7-day pause, the patients presented for cryosurgery (liquid nitrogen with 1mm diameter aperture, −196 °C, freezing duration of 10 s, at 2 cm from the lesion surface), immediately followed by a 5% 5FU chemowrap for 24 h. Then, 8 occlusive applications of 5% imiquimod cream were made, with one application every two days. Patients were instructed to apply imiquimod at night, followed by an occlusive bandage, and to wash off the cream in the morning. Imiquimod applications were followed by a four-week healing phase. The second session of cryosurgery was subsequently performed, immediately followed by a 5% 5FU chemowrap for 24 h. After another four-week healing phase, clinical evaluation and dermoscopy were performed, and CTr remission (tumor clearance) was evaluated. Tumor clearance after CTr was defined as the absence of evidence of residual tumor via digital dermoscopy (Heine Delta 20 T attached to a Nikon digital camera).

#### 2.2.2. Surgery

For the surgical subgroup, a standard scalpel elliptical excision with a minimum of 5 mm safety margins of clinically normal skin was performed in patients with a clinical and digital dermoscopy diagnosis of BCC, under a local anesthesia of lidocaine 1% and/or sedation. Subcutaneous and surface sutures were performed. The patients were reviewed at 10 to 14 days (depending on BCC localization) for the removal of sutures and a wound check. The diagnosis of BCC was histopathologically confirmed in all patients who underwent surgical excision or CTr.

#### 2.2.3. Adverse Events

Patients were questioned about local AE (pain and pruritus) at each medical visit, following the procedure. The levels of local pain and pruritus were divided based on severity (mild, moderate, and severe), in both subgroups. The MD who performed the procedure also evaluated potential local infection, bullae after cryosurgery, scaling, and surgical wound dehiscence. If present, they were reported separately from the pain and pruritus levels. Unexpected AE were also noted, if present. All of the AE were collected for each patient during the entire treatment protocol, and were considered in the final analysis.

#### 2.2.4. Patients Follow-Up

The follow-up was performed three months after clearance, and every three to six months afterward, regardless of the treatment group. Patients who did not present for the first three month evaluation were considered lost to follow-up.

For both treatment subgroups, follow-up examinations consisted of clinical, photographic, and digital dermoscopy examinations. Surgical margins were evaluated. The absence of any BCC dermoscopy criteria was considered sufficient to exclude relapse [14]. Follow-up examinations were extended for each patient with a telephone follow-up (TFU), three years after the clearance date. All of the examinations performed before TFU were performed in person. During TFU, signs and symptoms of recurrent BCC were questioned/recorded. The follow-up period (in-person and TFU) ended on 1 November 2021.

#### 2.2.5. Cosmetic Outcome

Scarring and pigmentation changes were considered when assessing the final cosmetic outcome. The authors considered a minimum/absent scarring as a good cosmetic result. Cosmetic self-assessment was achieved by asking patients to rate the cosmetic outcome (very satisfied/moderately/slightly/not at all) during TFU.

### 2.3. Statistical Analysis

All investigative data were collected into a central database (Microsoft Excel) by the study coordinator. The statistical analysis was completed using the R software. Descriptive and graphical analysis was used to check assumptions of normality and linearity for all study variables. The normality of the distributions was also tested rigorously by Shapiro–Wilk test, and the symmetry of non-normal distributions by looking at the skewness and kurtosis indicators.

Differences between subgroups were analyzed using an unpaired two-tailed *t*-test or Mann–Whitney U test for all continuous scale data between subgroups, where appropriate. A chi-square (χ^2^) or Fisher’s exact test were used, where appropriate, to compare the prevalence of comorbid risk and AE between treatment subgroups. They were also used in the subgroup analyses, where we assessed whether intervention effect/AE varied by sex, skin type, site of lesion, tumor size (<10 mm, 10–30 mm, ≥30 mm), and age groups (30–54, 55–74, >75); we also tested the independence of treatment subgroups with regards to sex, skin type, subtype, and site of the lesion. Linear relations with a *p*-value (two-sided) ≤ 0.05 were considered significant.

## 3. Results

### 3.1. Participants

A total of 179 patients were included in the study: 129 patients opted for conservative treatment, and 50 patients for radical surgery (Figure 2). Among the CTr patients, 61 (47.29%) patients chose this treatment because they had surgical anxiety (refusal of surgery), 19 (14.73%) patients had comorbidities (insulin-dependent diabetes, heart failure, anticoagulation, liver disease, history of neoplasia, etc.) and considered lower risks to be associated with CTr, and 49 (37.98%) patients considered themselves too old to undergo surgery. Additionally, most of the patients also considered CTr for potential cosmetic advantages. All of the histopathological examinations confirmed non-ulcerated BCCs. A total of 113 patients completed the full CTr regimen (87.6%), 16 abandoned the regimen (12.4%) and 10 were excluded from the final analyses due to a lack of follow-up. The patients abandoned the conservative treatment for the following reasons: considered it too prolonged and chose surgery instead; left the country; accessibility reasons. Additionally, some were discovered to have visceral neoplasia in the meantime, and underwent surgery and oncological treatment. As such, they did not present for CTr continuation. Therefore, 103 patients from the conservative subgroup were analyzed. From the surgical subgroup, 4 patients were lost to follow-up, and 46 surgical patients were analyzed. An overall number of 149 patients completed the procedures and were analyzed.

There were no significant differences between the two subgroups regarding gender, skin type, and BCC localizations. Subjects from the conservative subgroup were significantly older, and had a higher prevalence of comorbidities. The mean dimension of the BCCs was significantly lower in the CTr subgroup. The mean dimension of BCC varied between subgroups in patients with the histopathological nodular subtype, skin type II, and skin type III. Patients with tumor sizes below 10mm preferred the CTr to the surgical treatment, but those with BCCs between 10–30 mm preferred surgery instead (Table 1 and Table 2).

### 3.2. Outcome Data and Main Results

Of the 113 patients who underwent the full CTr regimen, 112 had remission after CTr (clearance rate: 99.11%); 102 patients were followed up, and 101 patients did not show BCC recurrence during the follow-up (recurrence rate: 0.98%). The mean duration of the CTr was 3.81 ± 1.48 months. The average duration of follow-up of patients in the conservative subgroup was 21.11 ± 14.57 months (CI: 18.25, 23.97); 8.93% were lost to follow-up, 22 (19.64%) were followed up for less than 12 months, 40 (35.71%) were followed up between 12 and 24 months, and 40 (35.71%) were followed up for more than 24 months (Figure 3).

An 82-year-old female patient with a 25 mm superficial BCC localized on her posterior thorax did not show clearance to CTr, and was switched to radiotherapy. A 58-year-old female patient with a 50 mm superficial pigmented BCC at diagnosis showed clinical and histopathological recurrence 48 months after CTr completion. The recurrent BCC was removed using a surgical punch excision.

All patients from the surgical group underwent successful surgery; 46 patients were followed up, and we did not encounter any recurrences in the surgical group during the follow-up. The average duration of the follow-up of surgically treated patients was 18 ± 9.77 months (CI: 15.09, 20.09); 8% were lost to follow-up, 6% were followed up for less than 12 months, 64% were followed up between 12 and 24 months, and 22% were followed up for more than 24 months.

TFU did not reveal new BCC recurrences in the two subgroups. We did not find statistical dependencies either between clearance rate and treatment (*p* = 1), or between recurrence rate and treatment (*p* = 1). Therefore, we did not find radical surgical excision to be superior to this conservative treatment in terms of clearance or recurrence rates. Furthermore, histopathological subtypes did not correlate with clearance or recurrence rates.

### 3.3. Adverse Events (AE)

A majority of 70.8% of patients from the CTr subgroup did not present any AE during treatment. Only 14.16% had mild and 15% had moderate local AE (pain and pruritus, especially after laser/cryosurgery and/or imiquimod). 5-FU chemowraps were well tolerated. No severe local or systemic AE were encountered in the CTr subgroup. In total, two patients had a secondary impetiginization that responded promptly to a topical antibiotic powder containing zinc bacitracin 250 UI and neomycin sulfate 5000 UI, which was administered for 5 days. One patient presented a bulla after cryosurgery that was punctured with a sterile needle. None of these three patients abandoned the CTr.

Within the surgical subgroup, 76% of patients did not present AE, 14% had mild, and 10% presented moderate local AE (after surgery or stitches removal). We did not encounter any wound dehiscence or wound impetiginization. In both subgroups, all of the reported AE were expected, and in a definite causal relationship with the treatment.

It is worth mentioning that we did not find any statistical difference between the two subgroups regarding the AE. There was no statistically significant difference in the distribution of AE according to gender for the two subgroups. Furthermore, we have analyzed AE according to age and tumor size, and proved that there was no statistically significant difference between the conservative and the surgical subgroup for either of these subgroups (Figure 4).

### 3.4. Cosmetic Outcome

Within the Ctr subgroup, 89.32% of the patients declared themselves very satisfied, and 10.68% were satisfied. Additionally, 50% of the patients from the surgical subgroup were very satisfied, 34.78% moderately, and 15.22% slightly satisfied. Therefore, the self-assessment cosmetic outcome revealed a superior (*p* < 0.001) cosmetic outcome in the Ctr subgroup. We consider the healing process was superior in the CTr group, as we did not encounter any hypertrophic/keloid scarring after cryosurgery or ablative laser. Overall, the pigmentation changes and visible scarring were significantly more prominent after standard surgery. 

## 4. Discussion

This work showed that elderly and frail patients with primary, non-ulcerated BCCs are more suitable for CTr over radical surgery. Adverse events, such as pain and pruritus, did not show a statistical significance between the two subgroups. The clearance rate and the recurrence rate were independent of the two treatments. A significantly better cosmetic outcome was displayed in the CTr subgroup, as compared to post-surgical scars; however, we are aware that patients with larger tumors are more often put forward for surgery. Both methods have advantages and disadvantages (Figure 5), as has been revealed to the authors during the monitoring of patients.

One of the strengths of our study is the large number of patients included in the CTr group. Another powerful feature is the inclusion of the so-called ‘high-risk’ histological subtypes, while the literature usually recommends non-surgical therapies for ‘low-risk’ BCC subtypes. Of note, in this study, a superficial BCC did not respond to CTr, and a superficial pigmented BCC recurred after CTr (both considered ‘low-risk’). The long follow-up period, completed with TFU, represents another strength of this study.

This multi-step conservative approach is based on the principle of synergistic potentiation; hence, we find it difficult to compare the singular applied steps to any other study, because studies comprising this kind of combination setting are lacking. Cryosurgery and topical therapies, such as imiquimod and 5 FU, can be used when alternatives to surgery are required [22]. Geisse et al. proved that the proportion of patients who showed remission after a 12-weeks twice daily 5% imiquimod monotherapy was 100%, higher than one daily group (87.1%), 5 times a week group (80.8%), and 3 times a week group (51.7%) [23]. In a study by Gross et al., BCCs treated with 5FU twice daily for 12 weeks showed a 90% cure rate [24]. Therefore, our clearance rate is comparable to the 12-week 5% twice daily imiquimod monotherapy treatment, but superior to the 12-week 5% twice daily 5FU monotherapy treatment. We assume that we managed to overcome the low 5-FU local penetration by applying it post-procedurally, in the form of a chemowrap. Both laser and cryosurgery may increase 5-FU local absorption. Of note, our CTr duration is comparable to the usual 12 weeks of local therapies with imiquimod or 5FU in monotherapy. Hextall et al. found that more BCC patients treated with topical imiquimod (monotherapy) develop AE, as compared to the surgical group [10]; however, in our study, there was no significant difference between the reported local AE from the conservative group and the surgical group, suggesting that this combined method might be a good alternative to surgery, including in patients who feel pain-related anxiety or surgical anxiety.

With cure rates exceeding 90%, cryosurgery is considered the most effective for low-risk BCCs involving the trunk and limbs [25]. Given the long freeze time of at least 30 s, a satisfying success rate using cryosurgery alone may be accompanied by significant unwanted side effects, such as hemorrhagic blistering, edema, hypopigmentation, and scarring [26]. Using a 10-s freeze cycle with this combination of topicals, we did not encounter severe AE, such as edema, scarring, or hypopigmentation. Furthermore, none of our patients abandoned the CTr due to adverse events.

A high recurrence rate may limit conservative treatments in BCC. For example, Gollnick et al. studied the recurrence rate of superficial BCC in 182 patients, following treatment with imiquimod 5% cream, once daily, 5 ×per week for 6 weeks, and 89.6% had no clinical evidence of the tumor at the 12-week post-treatment assessment. In the first 12 months of follow-up, 10 clinical recurrences were noted, and the statistical data predicted a long-term outcome [27]. Vidal et al. found that 76% of patients diagnosed with superficial, nodular, and infiltrative BCCs, treated with 24 doses of 5% imiquimod cream, showed no clinical sign of tumor at the 6-week post-treatment evaluation. There were no late relapses, and the 5-year recurrence was only 2% [28]. In a cryoimmunotherapy study, MacFarlane and Kader El Tal studied the efficiency of cryosurgery followed by 5% imiquimod, and obtained a 2% recurrence rate. The authors assumed that cryosurgery may damage the stratum corneum, thereby facilitating imiquimod penetration into the deep skin layers [29]. This point of view may be in line with our technique, since we have used in our method both cryosurgery and ablative CO2 laser before 5FU and imiquimod applications. Soong and Keeling found a 73% clinical cure rate at a 6-month follow-up appointment for the combination of cryosurgery and a 3-week course of 5% 5-fluorouracil in superficial BCC [30], which is inferior to our 3-months clearance rate. A 2021 study revealed an overall 3-month response of 84.2% with an Er:YAG ablative fractional laser (AFL)-primed MAL-PDT (Er:YAG AFL-PDT) treatment for nodular BCC, which is inferior to our results. The same study obtained higher recurrence rates (6.3%) at a shorter follow-up period (12 months) [31]. Genouw et al. found an exceptional efficacy (100%) and good to excellent aesthetic results when combining CO_2_ continuous laser with PDT for superficial BCCs in a small size pilot study with a 12-month follow-up [32]. Curettage followed by daily treatment for 6 to 10 weeks with imiquimod 5% shows optimistic results at a 3-month follow-up [33]; this setting may be of high interest when lasers are not available. Although occlusive imiquimod was not shown to be superior to non-occlusive imiquimod in superficial and nodular BCCs [34]; considering our clinical experience, we preferred the occlusion in this setting. However, occlusive imiquimod can be irritating, and patients should be appropriately advised.

Adherence to imiquimod and antibiotic powder local regimens could not be monitored by the authors, and could be a limitation of this study. Although the gold standard diagnosis of BCCs is the pathology report, dermoscopy was solely used to evaluate BCC clearance and recurrences. This non-invasive method of diagnosis eliminated the local adverse events of an alternate biopsy, such as inflammation and scarring. Moreover, a biopsy scar could have seriously affected the cosmetic outcome, in both subgroups. We do not consider that this decision is a source of study limitation, since a prospective study of 3500 BCCs showed that dermoscopy has a very high sensitivity (93.3%) and specificity (91.8%) for detecting all types of BCCs [25]. However, histological follow-up may be warranted in future studies regarding this method. As it does not represent a standard practice, tumoral depths were not recorded [35,36]. At a first glance, few settings are changed when compared to the established treatments, but we end up with promising results.

In summary, we conducted a prospective, comparative study that encompasses a novel conservative BCC treatment, and compares it to the standard surgical excision, in terms of clearance, recurrence rates, local adverse events, and cosmetic outcomes. This conservative approach may be more easily tolerated than cryosurgery or imiquimod alone, and has comparable efficacy to surgery. Our work addresses a relevant subject i.e., conservative treatment of BCCs when surgery might not be the best option or is rejected for different reasons. To our knowledge, this is the first conservative treatment to comprise combined CO_2_ laser, cryosurgery, and topical agents for the treatment of primary, non-ulcerated BCCs. We aim to design a series of prospective, randomized studies in which different combinations of this approach will be assigned to different treatment groups. Furthermore, we intend to develop similar comparative settings for easy-to-treat and stages IIA and IIB difficult-to-treat (DTT) BCCs, classified according to the novel operational staging system [19].

## Figures and Tables

**Figure 1 jcm-11-03439-f001:**
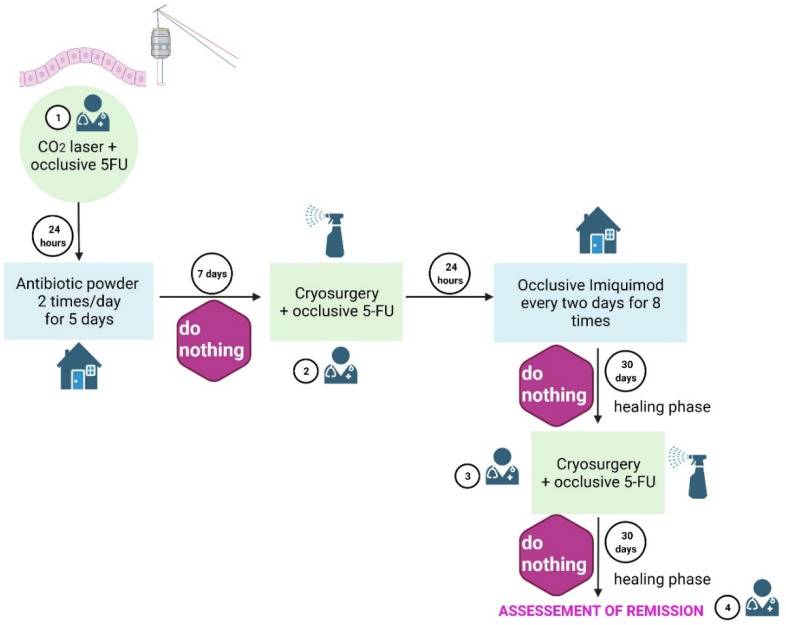
Conservative treatment methodology. This method comprises an initial CO_2_ laser ablation, followed by self-application of antibiotic powder, two cryosurgery sessions, three 5-FU chemowraps (after CO_2_ laser and cryosurgery), and eight occlusive self-applications of imiquimod. A total of four visits to the physician’s office (including the last one to establish BCC clearance) are required to complete the conservative treatment. 5-FU: 5-fluorouracil.

**Figure 2 jcm-11-03439-f002:**
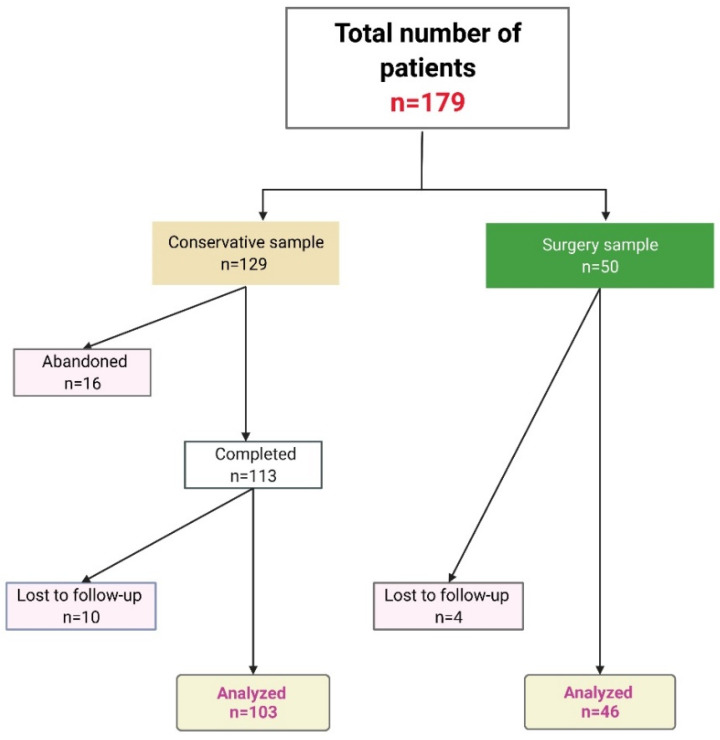
Subjects’ disposition in conservative and surgical subgroups.

**Figure 3 jcm-11-03439-f003:**
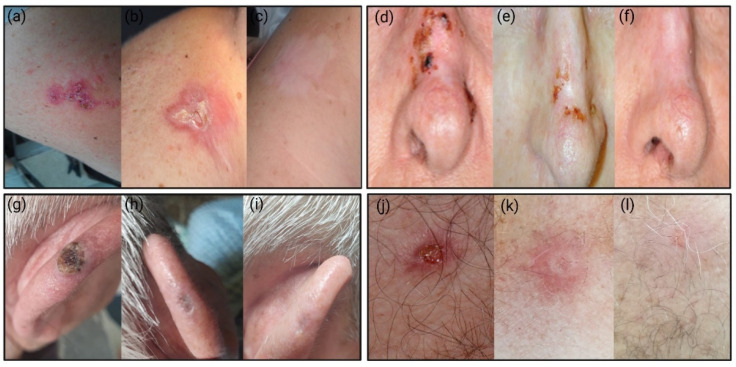
Initial (**a**,**d**,**g**,**j**), intermediate (**b**,**e**,**h**,**k**), and 12-month follow-up (**c**,**f**,**i**,**l**) clinical appearance in four patients with different BCC localizations (arm, nose, auricle, and anterior thorax). The 12-month evaluation displays satisfactory cosmetic outcomes.

**Figure 4 jcm-11-03439-f004:**
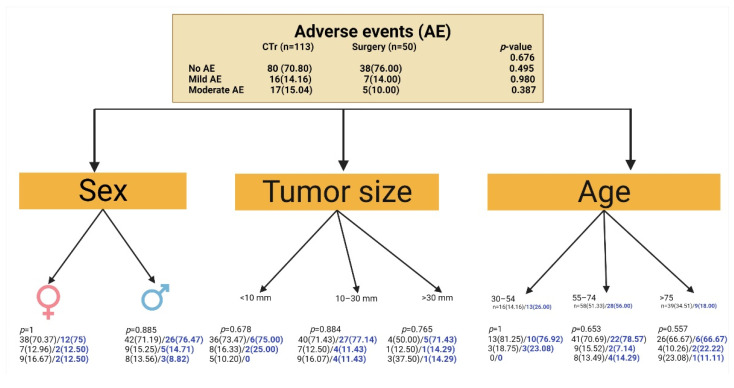
Adverse events (local pain and pruritus) experienced in treatment subgroups. The numbers in brackets represent the percentages with respect to the cardinal of the group. (Surgery with blue).

**Figure 5 jcm-11-03439-f005:**
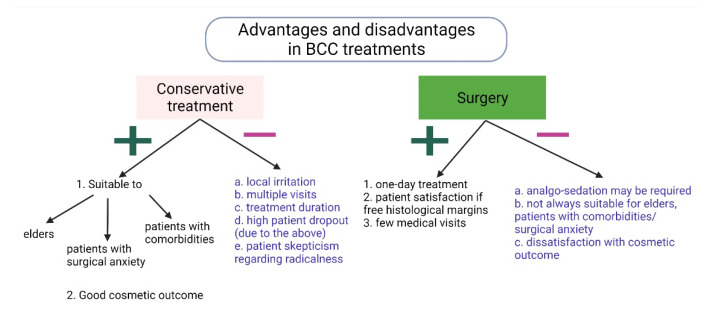
Advantages and disadvantages of conservative and surgical treatments for BCC.

**Table 1 jcm-11-03439-t001:** Demographic, clinical, and histopathological data.

		CTr (*n* = 129)	Surgery (*n* = 50)	*p*-Value
**Age**	Avg (SD)	67.78 (12.76)	61.64 (12.69)	0.007 ^1^
	CI (95%)	(65.56, 70.00)	(58.03, 65.25)	
	Median	77	63	
	Min	31	31	
	Max	91	82	
**Sex, *n* (%)**				0.064 ^2^
	M	68(52.71)	34(68)	
	F	61(47.29)	16(32)	
**Skin Type**				0.760 ^2^
	II	69(53.49)	28(56)	
	III	60(46.51)	22(44)	
**Subtype**				0.005 ^2^
	S	71(55.04)	14(28)	
	*n*	20(15.50)	11(22)	
	O	38(29.46)	25(50)	
**Age group**				0.029 ^2^
	30–54	16(12.40)	13(26)	
	55–74	70(54.26)	28(56)	
	>75	43(33.33)	9(18)	
**Comorbidities**	0	52(40.31)	34(68)	0.000 ^2^
	1	77(59.69)	16(32)	

^1^ Mann–Whitney U Test, ^2^ Chi-Squared Test. The numbers in brackets represent the percentages with respect to the cardinal of the group.

**Table 2 jcm-11-03439-t002:** Size variation and localizations in the two subgroups.

		CTr	Surgery	*p*-Value
**Size(mm)**	Avg (SD)	12.79 (9.71)	17.36 (10.62)	0.000 ^1^
	CI (95%)	(11.11, 14.49)	(14.34, 20.38)	
	Median	10	15	
	Min	1	4	
	Max	60	60	
	Skin Type = II	13.65	18.17	0.011 ^1^
	Skin Type = III	11.81	16.31	0.013 ^1^
	Subtype = *n*	11.35	20.45	0.008 ^1^
	Subtype = S	11.40	14.42	0.057 ^1^
	Subtype = O	16.15	17.64	0.337 ^1^
**Localization**				0.017 ^2^
	Body	51(39.53)	14(28)	
	Nose	36(27.91)	8(16)	
	Cheeks	18(13.95)	12(24)	
	Frontotemporal	9(6.98)	7(14)	
	Limbs	5(3.88)	6(12)	
	Ear	4(3.10)	0(0)	
	Scalp	4(3.10)	0(0)	
	Neck	2(1.55)	3(6)	

^1^ Mann–Whitney U Test, ^2^ Fisher’s Exact Test. The numbers in brackets represent the percentages with respect to the cardinal of the group.

## Data Availability

Not applicable.

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
