# Peer review of "A Conservative Combined Laser Cryoimmunotherapy Treatment vs. Surgical Excision for Basal Cell Carcinoma"

_jcm, 2022, doi:10.3390/jcm11123439_

Round 1
Reviewer 1 Report
This is an interesting study describing the efficacy of combined non-surgical methods for the treatment of BCC and compares it with radical excision.
# My main issue with this paper is the message that the “conservative” approach is more suitable for elderly and sicker patients. While both conservative tx and excision require local anesthesia, surgery requires 1-2 visits while the conservative option require multiple visits and much work from the patient’s side. Based on the results of the study I think this method should be offered to patients that are cosmetically concerned, but I do not see the advantage for elderly patients, especially when the average size of the tumors treated with the conservative approach was 13mm (I assume it is mm, please add the units to the table).
# Another major limitation of this paper is the lack of histology at follow-up. The authors mention that dermoscopy is an accurate method for diagnosing BCC, however most of the references to which they compare their results to relied on more accurate methods to validate complete response.
# Please better explain what part of the follow-up was done in-person and what part was telephone follow-up.
# Please mention which BCC subtypes were included as part of the “others”.
# Page 10 row 279- pigmented BCC can be any subtype, please mention which histopathilogical subtype.
#Figure 5 – Please add high patient dropout to the disadvantages of the conservative treatment
# Page 11 rows 305-306 – if patients did not abandon the CTr due to side effects – why did 16 patients abandon treatment?
Author Response
Response to Reviewer 1 Comments
Dear Reviewer,
Thank you very much for your extremely valuable remarks. We would like to emphasize that all your suggestions and comments have improved our manuscript.
Point 1: “My main issue with this paper is the message that the “conservative” approach is more suitable for elderly and sicker patients. While both conservative tx and excision require local anesthesia, surgery requires 1-2 visits while the conservative option require multiple visits and much work from the patient’s side. Based on the results of the study I think this method should be offered to patients that are cosmetically concerned, but I do not see the advantage for elderly patients, especially when the average size of the tumors treated with the conservative approach was 13mm (I assume it is mm, please add the units to the table).”
Response 1: When we refer to elderly and frail patients, we consider patients over 75-80 years old, with numerous comorbidities, often on anticoagulation or antiaggregant therapy. The oldest patient in the study is 91 years old. Most of these patients are accompanied by their close relatives, who refuse any conventional surgery. In addition, we find challenging for elderly patients with often consecutive multiple BCCs to perform so many excisions. When we refer to patients with comorbidities, we consider for example the case of an 85-year-old patient with acute lymphoblastic leukemia who had superficial BCC treated conservatively without recurrence, while surgery for the same patient for other superficial BCC had recurrence (the patient is a close relative the last author, not included in the study). Many patients had severe liver cirrhosis (with portal hypertension) and surgery could have triggered hepatic encephalopathy. Additionally, we had many patients with malignant or uncontrolled hypertension, and surgery may be a risk for these patients (increased bleeding may trigger hemodynamic consequences, including arrhythmias). Moreover, the literature cites local treatments in geriatric patients, but not in this combination (e.g.1. Sreekantaswamy S, Endo J, Chen A, Butler D, Morrison L, Linos E. Aging and the treatment of basal cell carcinoma. Clin Dermatol. 2019;37(4):373-378. doi:10.1016/j.clindermatol.2019.06.004. 2. ). This ("conservative") concept is therefore not contrary to the literature. We have added the unit of measurement (mm) to the table. Additionally, we have added (page 2): “Cosmetic concerns represent an attractive feature of conservative treatment, especially for less approachable areas (e.g. auricle).” We are thankful for this comment.
Point 2: “Another major limitation of this paper is the lack of histology at follow-up. The authors mention that dermoscopy is an accurate method for diagnosing BCC, however most of the references to which they compare their results to relied on more accurate methods to validate complete response.”
Response 2: Thank you for your comment. Dermoscopy is a highly efficient surface microscopy, that in trained hands has very high accuracy. The physicians’ authors are experts in dermoscopy and skin cancer (e.g., https://pubmed.ncbi.nlm.nih.gov/?term=olga%20simionescu&sort=date&sort_order=asc). It is widely used for the assessment of both melanoma and carcinoma (e.g 1. Ahnlide I, Zalaudek I, Nilsson F, Bjellerup M, Nielsen K. Preoperative prediction of histopathological outcome in basal cell carcinoma: flat surface and multiple small erosions predict superficial basal cell carcinoma in lighter skin types. Br J Dermatol. 2016 Oct;175(4):751-61; 2. Blum A, Hofmann-Wellenhof R, Marghoob AA, et al. Recurrent Melanocytic Nevi and Melanomas in Dermoscopy: Results of a Multicenter Study of the International Dermoscopy Society. JAMA Dermatol. 2014;150(2):138–145. doi:10.1001/jamadermatol.2013.6908). If the post excision scar does not reveal during digital dermoscopy (scroll x50) any features suggestive of a recurrence, or if following conservative treatment, no elements of dermoscopy recurrence/ subtotal clearance are noted, then taking a biopsy from a skin cancer free area would be unethical for the patient. Of course, the gold-standard diagnostic is the path report. However, the lack of a path report at follow-up with a performant digital dermoscopy examination (that excludes recurrence) fulfills the ethical approach of the patients and avoids unnecessary surgery. We have made adjustments on page 11.
Point 3: “Please better explain what part of the follow-up was done in person and what part was the telephone follow-up.”
Response 3: Thank you for this comment! All the follow-ups were performed in person, except the telephone follow-up which was not made in person (but by calling). We have explained, accordingly (page 5). We found this kind of follow-up extremely valuable, and we want to use TFU completed with video-call follow-up, especially for patients who live far from our hospital in our future research settings.
Point 4: “Please mention which BCC subtypes were included as part of the “others”. “
Response 4: Thank you for your comment! We have completed the BCC subtypes (O section).
Point 5: “Page 10 row 279- pigmented BCC can be any subtype, please mention which histopathological subtype.”
Response 5: Thank you for this comment! We have completed accordingly.
Point 6: “Figure 5 – Please add high patient dropout to the disadvantages of the conservative treatment”
Response 6: Thank you for this comment! We made the adjustments, according to Point 7, as well.
Point 7: “Page 11 rows 305-306 – if patients did not abandon the CTr due to side effects – why did 16 patients abandon treatment?”
Response 7: Thank you for this valuable observation! We have completed the abandon reasons in Section 3 (3.1-Participants): “The patients abandoned the conservative treatment for the following reasons: considered it too prolonged and chose surgery instead, left the country, accessibility reasons; additionally, some were discovered with visceral neoplasia in the meanwhile, underwent surgery and oncological treatment, and did not present for CTr continuation.” Furthermore, we have completed with a dropout percentage.
In the end, our research team would like to thank you for the hard work that you have put into evaluating our manuscript. We are aware that at this point our manuscript is highly improved.
With best wishes,
Olga Simionescu MD Ph.D., Clinical Professor and Chief, Department of Dermatology I
Carol Davila University of Medicine and Pharmacy, Bucharest, Romania
dana.simionescu@umfcd.ro
Reviewer 2 Report
This is an interesting study and I enjoyed reading it. I think that the method is interesting, however, the Authors seem to be overly optimistic about it and the conclusions need to be adjusted. I also enjoyed all the figures and diagrams a lot. Here are my specific comments:
1. It is unclear to me what the study period was. It is stated that the study period was 2013-2018. Is it including the follow-up period? If not - why you decided not to extend the follow-up period, and you are publishing the results 4 years after finishing the study.
2. The follow-up period is stated 21.11 +/- 14.57 months in the conservative group, while the usual follow-up period in the studies regarding reccurancies is 5 years.
3. I don't understand why the comparison was with the surgical treatment, not the destructive therapies. In the discussion, the authors should focus more on comparing their method with studies comparing destructive therapy or combined destructive and topical therapies.
- Choi SH, Kim KH, Song KH. Er:YAG ablative fractional laserprimed photodynamic therapy with methyl aminolevulinate as an alternative treatment option for patients with thin nodular basal cell carcinoma: 12-month follow-up results of a randomized, prospective, comparative trial.
- Genouw E, Verheire B, Ongenae K, De Schepper S, Creytens D, Verhaeghe E, et al. Laser-assisted photodynamic therapy for superficial basal cell carcinoma and Bowen’s Disease: a randomised intra-patient comparison between a continuous and a fractional ablative CO2 laser mode.
- Rhodes LE, de Rie MA, Leifsdottir R, Yu RC, Bachmann I, Goulden V, et al. Five year follow up of a randomized prospective trial of topical methyl aminolevulinate-photodynamic therapy versus surgery for nodular basal cell carcinoma.
- de Vijlder HC, Sterenborg HJ, Neumann HA, Robinson DJ, de Haas ER. Light fractionation significantly improves the response of superficial basal cell carcinoma to aminolaevulinic acid photodynamic therapy: five-year follow-up of a randomized, prospective trial.
- Roozeboom MH, Aardoom MA, Nelemans PJ, Thissen MR, Kelleners-Smeets NW, Kuijpers DI, et al. Fractionated 5- aminolevulinic acid photodynamic therapy after partial debulking versus surgical excision for nodular basal cell carcinoma: a randomized controlled trial with at least 5-year follow-up
- Christensen E, Mørk C, Skogvoll E. High and sustained efficacy after two sessions of topical 5-aminolaevulinic acid photodynamic therapy for basal cell carcinoma: a prospective, clinical and histological 10-year follow-up study.
- Smucler R, Vlk M. Combination of Er:YAG laser and photodynamic therapy in the treatment of nodular basal cell carcinoma.
- Wu JK, Oh C, Strutton G, Siller G. An open-label, pilot study examining the efficacy of curettage followed by imiquimod 5% cream for the treatment of primary nodular basal cell carcinoma
- Lucena SR, Salazar N, Gracia-Cazan˜a T, Zamarro´n A, Gonza´lez S, Juarranz A ´ , et al. Combined treatments with photodynamic therapy for non-melanoma skin cancer.
4. Although the study was done before, the current position statement of EADO regarding the classification of BCC should be mentioned in the introduction 10.1111/jdv.17467 as the current state of knowledge. If possible, the BCCs should be divided according to the current classification in both groups
5. As far as I understand the protocol mentioned in the study requires many visits in ambulatory care and good patient cooperation. Even though you have selected the participants to undergo this treatment, 16 have abandoned the therapy which is 12.4% - > this should be put in the results and discussed more. If these patients would have gone 1 day of surgical treatment the outcome would be, most probably, better.
6. Considering the cosmetic outcomes size of the tumor should be taken into consideration, as patients with bigger tumors were more often signed for surgical treatment.
7. I don't understand this paragraph An 82-years-old female patient with a 25 mm superficial BCC localized on her poste-219 rior thorax did not show clearance to CTr and was switched to radiotherapy. Another 58-220 years-old female patient with a 50 mm pigmentary BCC at diagnosis showed clinical and 221 histopathological recurrence 48 months after CTr completion. The recurrent BCC was re-222 moved using a surgical punch excision.
Please sum this up in the table.
8. The tables are not clear - what is in the ()?
In conclusion, the whole discussion part should be adjusted according to the results received and the limitations of the study.
Author Response
Response to Reviewer 2 Comments
Dear Reviewer,
Thank you very much for your extremely valuable remarks and your kind words. We would like to emphasize that all your suggestions and comments have improved our manuscript. We really feel optimistic in light of the results. These subgroups represent only a fraction of our experience with this method. We find it most important that dermatologists can recreate this procedure in ambulatory settings. Given your introduction comment, we have made adjustments (Introduction). Additionally, we have made important adjustments by responding to Q1-Q8. Thank you for your appreciation regarding our figures; they represent an original work done in BioRender, by Prof. Olga Simionescu.
Point 1: “It is unclear to me what the study period was. It is stated that the study period was 2013-2018. Is it including the follow-up period? If not - why you decided not to extend the follow-up period, and you are publishing the results 4 years after finishing the study.”
Response 1: Thank you for this remark, you are definitely right. This period (enrollment period, as specified in the manuscript) does not include the follow-up period: in-person follow-up (clinical, photographic, and digital dermoscopy examinations), followed by telephone follow-up (approximately 3 years after the clearance date, for each patient). The telephone follow-up was completed in November 2021 in all patients. We have made clarifications and added details on this matter in 2.2.4 section (Materials and Methods).
Point 2: “The follow-up period is stated 21.11 +/- 14.57 months in the conservative group, while the usual follow-up period in the studies regarding reccurancies is 5 years.”
Response 2: Thank you for this comment. The in-person follow-up was completed with telephone follow-up (TFU), which was made 3 years after the clearance date, in order to increase the observation period. In the statistics, we presented the in-person follow-up period, but it was prolonged with this TFU (we found this kind of follow-up extremely valuable, and we plan to use TFU completed with video-call follow-up, especially for patients who live far from our hospital in future studies). Of note, some high-rated papers in the literature comprising combined methods for BCC treatments used shorter follow-up periods (12-months) (e.g., Choi SH, Kim KH, Song KH. Er:YAG ablative fractional laser primed photodynamic therapy with methyl aminolevulinate as an alternative treatment option for patients with thin nodular basal cell carcinoma: 12-month follow-up results of a randomized, prospective, comparative trial. - Genouw E, Verheire B, Ongenae K, De Schepper S, Creytens D, Verhaeghe E, et al. Laser-assisted photodynamic therapy for superficial basal cell carcinoma and Bowen’s Disease: a randomised intra-patient comparison between a continuous and a fractional ablative CO2 laser mode.)
Point 3: “I don't understand why the comparison was with the surgical treatment, not the destructive therapies. In the discussion, the authors should focus more on comparing their method with studies comparing destructive therapy or combined destructive and topical therapies.
- Choi SH, Kim KH, Song KH. Er:YAG ablative fractional laserprimed photodynamic therapy with methyl aminolevulinate as an alternative treatment option for patients with thin nodular basal cell carcinoma: 12-month follow-up results of a randomized, prospective, comparative trial.
- Genouw E, Verheire B, Ongenae K, De Schepper S, Creytens D, Verhaeghe E, et al. Laser-assisted photodynamic therapy for superficial basal cell carcinoma and Bowen’s Disease: a randomised intra-patient comparison between a continuous and a fractional ablative CO2 laser mode.
- Rhodes LE, de Rie MA, Leifsdottir R, Yu RC, Bachmann I, Goulden V, et al. Five year follow up of a randomized prospective trial of topical methyl aminolevulinate-photodynamic therapy versus surgery for nodular basal cell carcinoma.
- de Vijlder HC, Sterenborg HJ, Neumann HA, Robinson DJ, de Haas ER. Light fractionation significantly improves the response of superficial basal cell carcinoma to aminolaevulinic acid photodynamic therapy: five-year follow-up of a randomized, prospective trial.
- Roozeboom MH, Aardoom MA, Nelemans PJ, Thissen MR, Kelleners-Smeets NW, Kuijpers DI, et al. Fractionated 5- aminolevulinic acid photodynamic therapy after partial debulking versus surgical excision for nodular basal cell carcinoma: a randomized controlled trial with at least 5-year follow-up
- Christensen E, Mørk C, Skogvoll E. High and sustained efficacy after two sessions of topical 5-aminolaevulinic acid photodynamic therapy for basal cell carcinoma: a prospective, clinical and histological 10-year follow-up study.
- Smucler R, Vlk M. Combination of Er:YAG laser and photodynamic therapy in the treatment of nodular basal cell carcinoma.
- Wu JK, Oh C, Strutton G, Siller G. An open-label, pilot study examining the efficacy of curettage followed by imiquimod 5% cream for the treatment of primary nodular basal cell carcinoma
- Lucena SR, Salazar N, Gracia-Cazan˜a T, Zamarro´n A, Gonza´lez S, Juarranz A ´ , et al. Combined treatments with photodynamic therapy for non-melanoma skin cancer.”
Response 3: Thank you very much for this valuable observation. First of all, the gold standard treatment is radical/ Mohs surgery according to the guidelines. Additionally, in Europe (particularly in Eastern Europe), there are huge centers of surgical dermatology. The majority of skin surface surgeries are performed by dermatologists (e.g., according to “Dermatologic Surgery”, I:”Fundamentals”, J.Kantor) and this idea represented our reference point, although literature presents some combined destructive combinatory techniques. We have referred in Discussions to some of these techniques (e.g., MacFarlane DF, El Tal AK. Cryoimmunotherapy: superficial basal cell cancer and squamous cell carcinoma in situ treated with liquid nitrogen followed by imiquimod. Arch Dermatol. 2011 Nov;147(11):1326-7. Soong LC, Keeling CP. Cryosurgery + 5% 5-Fluorouracil for Treatment of Superficial Basal Cell Carcinoma and Bowen's Disease [Formula: see text]. J Cutan Med Surg. 2018 Jul/Aug;22(4):400-4.) Additionally, we have now included in the Discussion section more comparisons with other techniques, as you recommended in this comment. Thank you for all the valuable papers that you proposed in order to strengthen our discussion section.
Point 4: “Although the study was done before, the current position statement of EADO regarding the classification of BCC should be mentioned in the introduction 10.1111/jdv.17467 as the current state of knowledge. If possible, the BCCs should be divided according to the current classification in both groups.”
Response 4: Thank you for this accurate update. According to this 2021 JEADV position statement, but also to other papers published in JAAD and BJD, it is of high importance to relate the clinical and dermoscopy features of BCC with the right treatment. Given the fact that this position statement was published in JEADV in late 2021, and we have developed this manuscript according to the inclusion criteria and to the Ethics Committee of Carol Davila University before this classification became available, this division according to the novel classification is not readily available in our setting, however, we have shared this novel state of knowledge in the Introduction and Discussion sections. We would like to emphasize that this modification is heavily improving our perspectives on this conservative treatment and we are grateful for your excellent point of view!
Point 5: “As far as I understand the protocol mentioned in the study requires many visits in ambulatory care and good patient cooperation. Even though you have selected the participants to undergo this treatment, 16 have abandoned the therapy which is 12.4% - > this should be put in the results and discussed more. If these patients would have gone 1 day of surgical treatment the outcome would be, most probably, better. “
Response 5: Yes, it is true. On the other hand, most of these patients are accompanied by their close relatives, who refuse any conventional surgery. Many patients had severe liver cirrhosis (with portal hypertension) and surgery could have triggered hepatic encephalopathy. Additionally, we had many patients with malignant or uncontrolled hypertension, and surgery may be a risk for these patients (increased bleeding may trigger hemodynamic consequences, including arrhythmias). We have completed the abandon reasons in Section 3 (3.1-Participants): “The patients abandoned the conservative treatment for the following reasons: considered it too prolonged and chose surgery instead, left the country, accessibility reasons; additionally, some were discovered with visceral neoplasia in the meanwhile, underwent surgery and oncological treatment, and did not present for CTr continuation.” Additionally, we have completed the results with the abandon rate and added to the Figure 5, at disadvantages, “high dropout rate”.
Point 6: “Considering the cosmetic outcomes size of the tumor should be taken into consideration, as patients with bigger tumors were more often signed for surgical treatment.”
Response 6: Thank you for this remark. During the clinical practice, it has been revealed to us that any scalpel incision will leave a more profound and less esthetical scar when compared to a CO2 laser. Your point of view is of high value and we have added it to the discussion section.
Point 7: “I don't understand this paragraph An 82-years-old female patient with a 25 mm superficial BCC localized on her posterior thorax did not show clearance to CTr and was switched to radiotherapy. Another 58-years-old female patient with a 50 mm pigmentary BCC at diagnosis showed clinical and 221 histopathological recurrence 48 months after CTr completion. The recurrent BCC was removed using a surgical punch excision. Please sum this up in the table. “
Response 7: Given the fact that we had one patient who did not show clearance to the CTr and one patient who developed a recurrence 2 years after the Ctr completion we have decided to detail these 2 cases broadly. The non-responder patient was switched to radiotherapy and the recurrent BCC was surgically removed. We made some adjustments to make this paragraph more precise. Thank you for this comment!
Point 8: The tables are not clear - what is in the ()?
Response 8: The numbers in brackets represent the percentages with respect to the cardinal of the group. We made this clarification under each table. Thank you for this comment.
In the end, our research team would like to thank you for the hard work that you have put into evaluating our manuscript. We are aware that at this point our manuscript is highly improved.
With best wishes,
Olga Simionescu MD Ph.D., Clinical Professor and Chief, Department of Dermatology I
Carol Davila University of Medicine and Pharmacy, Bucharest, Romania
dana.simionescu@umfcd.ro

Round 2
Reviewer 1 Report
I would like to thank the authors for their revisions. I find this manuscript now suitable for publication.
Reviewer 2 Report
All my comments have been addressed.